# Beyond GNNs: A Sample-Efficient Architecture for Graph Problems

## Abstract

Despite their popularity in learning problems over graph structured data, existing *Graph Neural Networks* (GNNs) have inherent limitations for fundamental graph problems such as shortest paths, $k$-connectivity, minimum spanning tree and minimum cuts. In all these instances, it is known that one needs GNNs of high depth, scaling at a polynomial rate with the number of nodes $n$, to provably encode the solution space. This in turn affects their statistical efficiency thus requiring a significant amount of training data in order to obtain networks with good performance. In this work we propose a new hybrid architecture to overcome this limitation. Our proposed architecture that we call as $GNN^+$ networks involve a combination of multiple parallel low depth GNNs along with simple pooling layers involving low depth fully connected networks. We provably demonstrate that for many graph problems, the solution space can be encoded by $GNN^+$ networks using depth that scales only *poly-logarithmically* in the number of nodes. This significantly improves the amount of training data needed that we establish via improved generalization bounds. Finally, we empirically demonstrate the effectiveness of our proposed architecture for a variety of graph problems.

## 1 Introduction

In recent years Graph Neural Networks (GNNs) have become the predominant paradigm for learning problems over graph structured data (Hamilton et al., 2017; Kipf & Welling, 2016; Veličković et al., 2017). Computation in GNNs is performed by each node sending and receiving messages along the edges of the graph, and aggregating messages from its neighbors to update its own embedding vector. After a few rounds of message passing, the computed node embeddings are aggregated to compute the final output (Gilmer et al., 2017). The analogy to message passing leads to a simple and elegant architecture for learning functions on graphs. On the other hand, from a theoretical and practical perspective, we also need these architectures to be *sample efficient*, i.e., learnable from a small number of training examples, where each training example corresponds to a graph. Recent works have shown that generalization in GNNs depends upon the depth of the architecture, i.e., the number of rounds of message passing, as well as the embedding size for each node in the graph (Garg et al., 2020). However, this requirement is in fundamental conflict with the message passing framework. In particular, using GNNs to compute several fundamental graph problems such as *shortest paths, minimum spanning tree, min cut* etc., necessarily requires the product of the depth of the GNN and the embedding size to scale as $\sqrt{n}$ where $n$ is the size of the graph (Loukas, 2020). This in turn places a significant statistical burden when learning these fundamental problems on large scale graphs. The above raises the the following question: *Can one develop sample efficient architectures for graph problems while retaining the simplicity of the message passing framework?*

Several recent works have tried to address the above question by proposing extensions to the basic GNN framework by augmenting various pooling operations in conjunction with message passing rounds to capture more global structure (Ying et al., 2018; Simonovsky & Komodakis, 2017; Fey et al., 2018). While these works demonstrate an empirical advantage over GNNs, we currently do not know of a general neural architecture that is versatile enough to *provably* encode the solution space of a variety of graph problems such as shortest paths and minimum spanning trees, while being significantly superior to GNNs in terms of statistical efficiency. In this work we propose a theoretically principled architecture, called $GNN^+$ networks for learning graph problems. While the basic GNN framework is inspired from classical message passing style models studied in distributed computing,

we borrow from two fundamental paradigms in graph algorithm design namely, sketching and parallel computation, to design GNN$^+$ networks. As a result of combining these two powerful paradigms, we get a new neural architecture that simultaneously achieve low depth and low embedding size for many fundamental graph problems. As a result our proposed GNN$^+$ architecture have a significantly smaller number of parameters that provably leads to better statistical efficiency than GNNs. Before we present our improved architecture, we briefly describe the standard GNN framework.

**Model for GNNs.** In this work we will study GNNs that fall within the message passing framework and using notation from previous works we denote such networks as GNN$^{mp}$ (Loukas, 2020). A GNN$^{mp}$ network operates in the AGGREGATE and COMBINE model (Gilmer et al., 2017) that captures many popular variants such as GraphSAGE, Graph Convolutional Networks (GCNs) and GIN networks (Hamilton et al., 2017; Kipf & Welling, 2016; Xu et al., 2019a). Given a graph $G = (V, E)$, let $x_i^{(k)}$ denote the feature representation of node $i$ at layer $k$. Then we have

$$a_i^{(k-1)} = \text{AGGREGATE}(\{x_j^{(k-1)} : j \in N(i)\}) \tag{1}$$

$$x_i^{(k)} = \text{COMBINE}(x_i^{(k-1)}, a_i^{(k-1)}). \tag{2}$$

Here $N(i)$ is the set of neighbors for node $i$. Typically the aggregation and combination is performed via simple one or two layer full connected networks (FNNs), also known as multi layer perceptrons (MLPs). In the rest of the paper we will use the two terms interchangeably.

**GNN$^+$ Networks.** Our proposed GNN$^+$ networks consist of one or more layers of a GNN$^+$ block shown in Figure 1. The GNN$^+$ block comprises of $r$ parallel GNN$^{mp}$ networks follows by $s$ parallel fully connected network modules for pooling where $r$ and $s$ are the hyperparameters of the architecture. More importantly we restrict the $r$ GNN$^{mp}$ modules to share the same set of weights. Hence the parallel GNN$^{mp}$ modules only differ in the way the node embeddings are initialized. Furthermore, we restrict each GNN$^{mp}$ to be of low depth. In particular, for degree-$d$ graphs of diameter $D$, over $n$ nodes, we will restrict the GNN$^{mp}$ to be of depth $O((d + D) \cdot \text{polylog}(n))$. Similarly, we require the $s$ fully connected networks to be of depth $O((d + D) \cdot \text{polylog}(n))$ and share the network weights. We connect the outputs of the GNN$^{mp}$ modules to the fully connected pooling networks in a sparse manner and restrict the input size of each fully connected network to be $O((d + D) \cdot \text{polylog}(n))$. Stacking up $L$ layers of GNN$^+$ blocks results in a GNN$^+$ network that is highly parameter efficient and in total has $O((d + D)L \cdot \text{polylog}(n))$ parameters. For such a network we call the depth as the total number of message passing rounds and the number of MLP layers used across all the $L$ stacks. Since we restrict our MLPs and GNN$^{mp}$ blocks inside a GNN$^+$ network to be of low depth, we will often abuse notation and refer to a GNN$^+$ architecture with $L$ stacks of GNN$^+$ blocks as a depth $L$ architecture. Our proposed design lets us alternate between local computations involving multiple parallel GNN blocks and global post-processing stages, while still being sample efficient due to the enforced parameter sharing. We will show via several applications that optimal or near-optimal solutions to many popular graph problems can indeed be computed via a GNN$^+$ architecture. Below we briefly summarize our main results.

**Overview of Results.** To demonstrate the generality of our proposed GNN$^+$ architecture, we study several fundamental graph problems and show how to construct efficient GNN$^+$ networks to compute optimal or near optimal solutions to these problems. In particular, we will focus on degree-$d$ graphs, i.e., graphs of maximum node degree $d$, with $n$ nodes and diameter $D$ and will construct GNN$^+$ networks of depth $\text{polylog}(n)$ and $O\big((D + d)\text{polylog}(n)\big)$ total parameters.

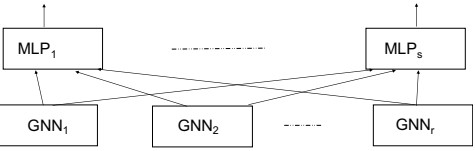

Figure 1: The basic GNN$^+$ block.

**Shortest Paths.** The first problem we consider is the fundamental graph problem of computing (approximate) all pairs shortest paths in undirected graphs. Given a graph $G = (V, E)$, let $d_G(u, v)$ be the shortest path between nodes $u$ and $v$. We say that an output $\{\tilde{d}_G(u, v) : u, v \in V\}$ is an

$\alpha$-approximate solution if for all $u \neq v$ it holds that

$$d_G(u,v) \leq \tilde{d}_G(u,v) \leq \alpha d_G(u,v).$$

We construct efficient GNN$^+$ networks for all pairs shortest paths with the following guarantee.

**Theorem 1** (Informal Theorem). *For any constant $c > 1$, there is a depth $O(D \log d + \log n)$ GNN$^+$ network with $O\big((n^{\frac{2}{c}} + d)\text{polylog}(n)\big)$ parameters that computes $(4c - 2)$-approximate all pairs shortest paths in the undirected unweighted degree-$d$ graphs over $n$ nodes. On the other hand, computing a $c$-approximate shortest paths using GNN$^{mp}$ networks requires a network of depth $\Omega(n)$.*

From the above theorem we see that by setting $c = O(\log n)$ we can encode a $c$-approximate solution using an $O(D \log d + \log n)$ GNN$^+$ network with only $O(d \cdot \text{polylog}(n))$ parameters. This is in stark contrast with the depth requirement of $\Omega(n)$ for the traditional GNN$^{mp}$ networks.

**Connectivity Measures.** Next we consider computing various graph connectivity measures. We first study the popular measure based on graph effective resistances (Chandra et al., 1996).

**Definition 1** (Effective Resistance). *Let $G$ be a weighted undirected graph $G$ with adjacency matrix $A$ and the associated Laplacian $L = D - A$. Given an edge $u, v$, the effective resistance between $u, v$ is defined as*

$$R_{u,v} = \xi_{u,v}^\top L^\dagger \xi_{u,v}.$$

*Here $\xi_{u,v}$ is an $n$ dimensional vector with $+1$ at position $u$, $-1$ at position $v$ and zeros everywhere. $L^\dagger$ refers to the matrix pseudo-inverse.*

We also study the following connectivity measure that was proposed by Panigrahy et al. (2012) in the context of web graphs. Given an undirected graph $G$, let $G_p$ be the random graph obtained by sampling each edge with probability $p$.

**Definition 2** (Affinity). *For any two vertices $u, v$ and for $p \in [0, 1]$, define $A_p(u, v)$ to be the probability that $u, v$ are connected in $G_p$. Then the affinity between $u$ and $v$ is defined as*

$$A(u,v) = \mathbb{E}_p[A_p(u,v)]$$

*where the expectation is taken over $p$ drawn from the uniform distribution in $[0, 1]$.*

For the above measures we show the following

**Theorem 2** (Informal Theorem). *There exists a GNN$^+$ architecture with $O(D \log(nd))$ parameters, and depth $O(D \log(nd))$ on graphs of diameter $D$ with $n$ nodes and maximum degree $d$, that approximate the above connectivity measures up to constant factors. On the other hand using GNN$^{mp}$ networks to compute the above measures, even approximately, necessarily requires a network of depth $\Omega(\sqrt{n})$.*

**Clustering, Minimum Cuts and Minimum Spanning Trees.** Finally, we showcase the power of a GNN$^+$ architecture for computing other fundamental graph problems. Given an undirected graph $G$, the spectral clustering of $G$ corresponds to the cut obtained by taking the sign of the eigenvector $v$ corresponding to the second smallest eigenvalue $\lambda_2(L)$, where $L$ is the graph Laplacian. For computing the spectral clustering via GNN$^+$ networks we show the following

**Theorem 3** (Informal Theorem). *There is a GNN$^+$ network of depth $\ell = O(\frac{1}{\lambda_2(L)\epsilon^2} \log n)$, with $O(d)$ parameters that computes an $\epsilon$-approximate spectral clustering on graphs of degree $d$. On the other hand, using GNN$^{mp}$ networks to even approximately compute the spectral clustering requires depth $\Omega(\sqrt{n})$.*

Next we consider the classical problems of computing a global minimum cut and minimum spanning trees in undirected graphs.

**Theorem 4** (Informal Theorem). *There exist GNN$^+$ networks of of depth $O((D + \log n) \log n)$, and $O(d)$ parameters for computing a global minimum cut (MINCUT ) and minimum spanning tree (MST) in degree $d$ graphs of diameter $D$. Furthermore, using GNN$^{mp}$ networks to compute these primitives (even approximately) necessarily requires depth $\Omega(\sqrt{n})$.*

**Generalization Bounds.** Our final result concerns the generalization properties of a depth $L$ $\text{GNN}^+$ architecture. For ease of exposition, we state here the results for the case when the $\text{GNN}^+$ architecture produces a one dimensional output. More general results are presented in Appendix D. Our generalization bounds depend on the depth $L$ and the total number of parameters $P$ in the $\text{GNN}^+$ network. Following recent work on providing generalization bounds for fully connected and convolutional neural networks (Bartlett et al., 2017; Long & Sedghi, 2020) that are based on *distance to initialization*, we consider the class $\mathcal{F}_\beta$ of depth $L$ $\text{GNN}^+$ networks with $P$ parameters that are at a distance $\beta$ from a reference parameter configuration (typically the parameters at random initialization). Let $y \in \mathbb{R}$ denote the output of the network and consider a Lipschitz loss function $\ell(y, \hat{y})$. Then, we provide following guarantee.

**Theorem 5** (Informal Theorem). *Let $\ell(\hat{y}, y)$ be a Lipschitz loss function bounded in $[0, B]$. Then, given $m$ i.i.d. samples $(G_1, y_1), (G_2, y_2), \ldots (G_m, y_m)$ generated from a distribution $D$, with probability at least $2/3$, it holds that for all $f \in \mathcal{F}_\beta$,*

$$\left| \hat{\mathbb{E}}_D[\ell_f] - \mathbb{E}_D[\ell_f] \right| \leq O\left( B\sqrt{\frac{P(\beta + L)}{m}} \right).$$

We refer the reader to Theorem 16 in Appendix D for a formal statement and the proof. Notice that the above theorem implies that our proposed $\text{GNN}^+$ architecture for the above graph problems can indeed be trained using very few samples as opposed to the traditional $\text{GNN}^{\text{mp}}$ networks since the $\text{GNN}^+$ network requires much fewer parameters and depth. Furthermore, since a $\text{GNN}^{\text{mp}}$ network is a special case of a $\text{GNN}^+$ architecture, our analysis also leads to an improved bound on the generalization guarantees for $\text{GNN}^{\text{mp}}$ networks as well. In particular, the above theorem improves upon the recent work of Garg et al. (2020) that provides generalization guarantees for training GNNs that scale with the branching factor of the graph. Using our improved analysis we are able to remove this dependence on the branching factor. See Appendix D for details.

## 2 RELATED WORK

GNNs operate primarily in the message passing framework where nodes aggregate and combine messages from their neighbors to update their embeddings. Several variants of this basic paradigm have been proposed, with each differing in how the aggregation and combination is performed. Popular variants include GraphSAGE (Hamilton et al., 2017), Graph Convolutions Networks (Kipf & Welling, 2016), GIN networks (Xu et al., 2019a), and graph pooling (Ying et al., 2018).

Various recent works have also studied the representation power of GNNs. The work of Xu et al. (2019a) demonstrates that the GNNs as considered in equation 1 are as powerful as the Weisfeiler-Lehman test for graph isomorphism (Weisfeiler & Lehman, 1968). The recent work of Xu et al. (2019b) compares the message passing framework of GNNs in representing computations involving dynamic programming. GNN networks that can capture higher order variants of the WL test have also been proposed recently (Maron et al., 2019).

Several works have also explored the limitations of GNNs for computing graph primitives. The work of Loukas (2020) established a correspondence between the message passing GNN framework and the well studied CONGEST model of distributed computing (Peleg, 2000). Based on the above correspondence it follows that in order to represent several important graph problems such as shortest paths, minimum cuts and minimum spanning tree, either the depth of the GNN or the embedding size of the nodes has to scale with the graph size at a polynomial rate. Notice that these lower bounds apply to any form of message passing framework and as a result recent work in incorporating non-symmetric node messages (Sato et al., 2019) in GNNs also run into the same barriers.

In order to address the above limitations recent works have proposed combining the GNN architecture with pooling mechanisms for aggregating more global information (Ying et al., 2018; Defferrard et al., 2016; Simonovsky & Komodakis, 2017; Fey et al., 2018; Bianchi et al., 2019; Du et al., 2019). For example the work of Ying et al. (2018) proposes a hierarchical approach where a GNN network is followed by a clustering step to compute higher level "nodes" to be used in the subsequent GNN operation. While these approaches show empirical promise, ours is the first work to design a principled architecture with theoretical guarantees that merges local distributed computations with global postprocessing stages.

Finally, the question of generalization for GNNs has also been studied in recent works. The most relevant to us is the recent work of Garg et al. (2020) that analyzes the Rademacher complexity of GNNs with the aggregate mechanism being a simple addition and the combine mechanism being a one layer neural network. Via analyzing the Rademacher complexity the authors show that the generalization for GNNs depends on the depth, the embedding size and the branching factor of the graph. Our improved analysis in Section D extends the result of Garg et al. (2020). Not only does our generalization bound apply to the more general GNN$^+$ networks, for the case of GNNs considered in (Garg et al., 2020) our analysis shows that the dependence on the branching factor can be eliminated in the generalization bounds. Generalization bounds have also been proved recently for GNN based networks that use the Neural Tangent Kernel (NTK) during the aggregation and combination operations (Du et al., 2019).

## 3    SHORTEST PATHS

In this section we provide a proof sketch of Theorem 1 showing how to construct an efficient GNN$^+$ architecture for the Shortest Paths problem. In particular we study all pairs shortest paths.

**All Pairs Shortest Paths.** The input is a graph $G = (V, E)$ with $n$ nodes. The desired output is an $\binom{n}{2}$ dimensional vector containing (approximate) shortest path values between each pair of vertices. Given a graph $G$, let $d_G(u, v)$ be the shortest path between nodes $u$ and $v$. We say that an output $\{\tilde{d}_G(u, v) : u, v \in V\}$ is an $\alpha$-approximate solution if for all $u \neq v$ it holds that

$$d_G(u, v) \leq \tilde{d}_G(u, v) \leq \alpha d_G(u, v).$$

We first show that the GNN$^{\mathrm{mp}}$ networks are highly inefficient for learning this problem.

**Theorem 6.** *Consider a GNN$^{\mathrm{mp}}$ $\mathcal{N}$ of depth $L$ over $n$ nodes where each node has a representation size of $B$. If $\mathcal{N}$ encodes $\alpha$-approximate all pairs shortest paths for graphs of diameter bounded by $D$, and for $\alpha < 3/2$, then it must hold that $B \cdot L \geq \Omega(n)$. Furthermore, for any GNN$^{\mathrm{mp}}$ that encodes $\alpha(n)$-approximate all pairs shortest paths it must hold that $B \cdot L \geq \Omega\big(\frac{n}{\alpha(n) \log n}\big)$. The lower bound holds for undirected unweighted graphs as well.*

*Proof.* The recent work of Loukas (2020) established that computation in GNN$^{\mathrm{mp}}$ networks is equivalent to the CONGEST model of computation popularly studied in the design of distributed algorithms (Peleg, 2000). In particular, a lower bound on the product of depth ($L$) and representation size ($B$) can be obtained by establishing the corresponding lower bound on the product of the number of rounds and the size of messages in the CONGEST model of computing. Furthermore, the result of Holzer & Wattenhofer (2012) shows that in the CONGEST model approximating all pairs shortest paths, even on unweighted undirected graphs requires the product of the number of rounds and the message size to be $\Omega(n)$. This was improved in the work of Nanongkai (2014) to show that for any $\alpha(n)$-approximation, the product of the number of rounds and the message size to be $\Omega\big(\frac{n}{\alpha(n) \log n}\big)$. Hence the corresponding lower bound on $B \cdot L$ follows.                    □

**Circumventing Lower Bounds via GNN$^+$.**    Next we detail our proposed GNN$^+$ architecture that can encode approximate shortest paths with significantly smaller depth and parameter requirements.

**Unweighted Graphs.** To illustrate the main ideas we study the case of undirected unweighted graphs. See Appendix A for the more general case of weighted graphs. The starting point of our construction is the following fundamental theorem of Bourgain (1985) regarding metric embeddings.

**Theorem 7** ((Bourgain, 1985)). *Any $n$-point metric $(X, d)$ can be embedded into the Euclidean metric of dimensionality $O(\log n)$ and distortion $O(\log n)$.*

The above theorem suggests that in principle, if we only want to estimate shortest paths up to an approximation of $O(\log n)$, then we only need node embeddings of size $O(\log n)$. If there were a GNN$^{\mathrm{mp}}$ network that could produce such embeddings, then one could simply compute the Euclidean distance between each pair of points to get the approximate shortest path. Furthermore, computing the Euclidean distance given the node embeddings can be done easily via a low depth full connected network. Unfortunately, producing the necessary low dimensional embeddings is exactly the task for

which GNN$^{\text{mp}}$ networks require large depth as proved in Theorem 6 above. While there do exist semi-definite programming based algorithms (Linial et al., 1995) for computing the embeddings required for Bourgain's theorem, they are not suitable for implementation via efficient neural architectures. Instead we rely on sketching based algorithms for computing shortest path distances.

In particular, for the unweighted case we adapt the sketch based approximate shortest path algorithms of Das Sarma et al. (2010) for designing an efficient network architecture. The sketch proposed in the work of Das Sarma et al. (2010) computes, for each node $u$, the distance of $u$ from a random subset $S$ of the nodes. This can be done via a simple breadth first search (BFS). Repeating this process $k$-times provides a $k$-dimensional embedding for each vertex and for an appropriate choice of $k$, these embeddings can be used to compute approximate shortest paths. Notice that this sketch based procedure is highly amenable to be implemented in a message passing framework. Overall, the algorithm performs multiple parallel BFS subroutines to compute the embeddings. It is also well known that BFS on diameter $D$ can be implemented by a GNN$^{\text{mp}}$ of depth $O(D)$.

Based on the above intuition, our proposed architecture is shown in Figure 2. It consists of $k$ parallel breadth first search (BFS) modules for $k = \Theta(n^{\frac{1}{c}} \log n)$ for a constant $c > 1$. Module $i$ computes the shortest path from each vertex in $G$ to any vertex in the set $S_i$. The sets $S_1, S_2, \ldots, S_k$ are randomly chosen subsets of the vertex set $V$ of various sizes. In particular there are $\Theta(n^{\frac{1}{c}})$ subsets of size 1, $\Theta(n^{\frac{1}{c}})$ subsets of size 2, $\Theta(n^{\frac{1}{c}})$ subsets of size $2^2$, and so on up to $\Theta(n^{\frac{1}{c}})$ subsets of size $2^{\lfloor \log n \rfloor}$. The BFS module $i$ produces $n$ distance values $v_1^{(i)}, \ldots, v_n^{(i)}$. These modules are followed by $\binom{n}{2}$ fully connected networks where each module is responsible for computing the approximate shortest path distance between a pair of vertices. In particular we have $\tilde{d}_G(s, t) = \max_i |v_s^{(i)} - v_t^{(i)}|$.

Notice from the discussion in Section 1 that the architecture in Figure 2 is a GNN$^+$ network with a single GNN$^+$ block. In the next section we will show how we can generalize to a suite of graph problems by stacking up multiple GNN$^+$ blocks. For our proposed network we have the following guarantee.

**Theorem 8.** *For any integer $c > 1$, and for a fixed graph topology over $n$ nodes with maximum degree $d$ and diameter $D$, there exists a neural network as shown in Figure 2 of size $O(n^{2+1/c})$, $\tilde{O}(n^{\frac{2}{c}})$ parameters, and depth $O(D \log d + \log n)$, that encodes $(2c - 1)$-approximate all pairs shortest paths in $G$.*

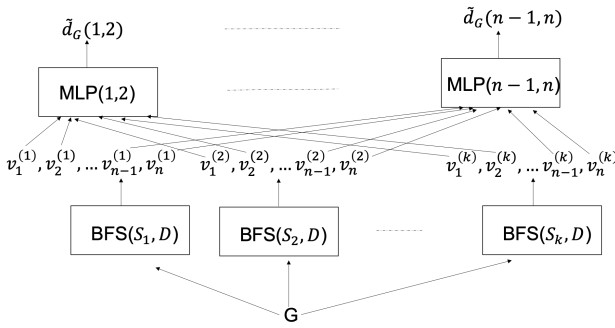

Figure 2: The network architecture for approximate all pairs shortest paths in unweighted graphs.

Before proving the theorem above we establish two supporting lemmas concerning the implementation of the BFS modules and the MLP module in the network architecture described in Figure 2.

**Lemma 1.** *The BFS module in Figure 2 can be implemented by a GNN of depth $O(D)$, $O(1)$ total parameters and with each node having a representation size of $O(1)$.*

**Lemma 2.** *For any $k$, the MLP module in Figure 2 can be implemented by a network of depth $O(\log k)$, $O(k^2)$ total parameters.*

*Proof of Theorem 8.* The correctness of the network architecture follows from the work of Das Sarma et al. (2010). Next we establish bounds on the total depth, size and the number of parameters. We have $k = \Theta(n^{\frac{1}{c}} \log n)$ copies of the BFS module. Each BFS module is of size $O(nd \log d)$ since there are $n$ nodes and each node implements a min function of size $O(d \log d)$. Hence, in total the BFS modules have size $O(n^{1+1/c} d \log d \log n)$. Next we have $\binom{n}{2}$ MLP modules each of size $O(k \log k)$ for a total size of $O(n^{2+1/c} \log n)$. Hence the total size of the neural network is bounded by $O(n^{2+1/c} \log n)$.

Next we bound the depth and the total number of parameters. The BFS module has $O(D)$ rounds with each requiring a depth of $O(\log d)$ for a total depth of $O(D \log d)$. The MLP module has a depth bounded by $O(\log k) = O(\log n)$. Hence the total depth is $O(D \log D + \log n)$. Finally, the BFS module requires $O(1)$ parameters and the MLP module requires $O(k^2)$ parameters. Hence, the total number of parameters in our architecture are bounded by $O(k^2) = O(n^{2/c})$. □

## 4 Minimum Cuts

To illustrate another application, in this section we design an efficient GNN$^+$ based architecture for computing the minimum cut in an undirected graph. We first argue in Appendix C that even computing an approximate mincut using traditional GNN$^{\mathrm{mp}}$ networks requires $\Omega(\sqrt{n})$ rounds. Our efficient GNN$^+$ based architecture is based on the parallel algorithm for computing mincut (Karger & Stein, 1996) and is shown in Figure 3. More importantly the architecture comprises of multiple layers of GNN$^+$ blocks in contrast to a single GNN$^+$ block in the case of shortest paths. The algorithm of Karger & Stein (1996) relies on the following lemma.

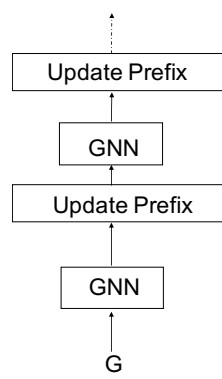

**Lemma 3** ((Karger & Stein, 1996)). *Let $G = (V, E)$ be an undirected unweighted graph with $m$ edges and $n$ vertices. Then with probability at least $\frac{1}{n^2}$, a random ordering $L$ of the edges contains a prefix $L'$ of $L$ such that the graph $G' = (V, L')$ contains exactly two connected components corresponding to the global minimum cut in the graph.*

Using the above, Karger & Stein (1996) proposed a Monte-Carlo randomized algorithm to compute the global minimum cut. For a given ordering $L$, the algorithm estimates the length of the prefix $L'$ corresponding to the cut by using a binary search procedure. This provides the set of *active edges*, i.e., edges in $L'$. Then one can run a connected components algorithm using edges in $L'$. If the prefix is too small, it results in more than two connected components; if it is too large it produces one connected component. If the number of connected components is two then the algorithm stops. Otherwise it recurses on the appropriate side of $L'$.

Figure 3: The network architecture for minimum cut.

We can implement the above algorithm using the GNN$^+$ architecture of depth $O(\log m)$ where the computation in each pair of (GNN,Update Prefix) blocks corresponds to executing one call of the above binary search procedure. During each call one needs to perform a connected component subroutine. This can be done via BFS and is implemented in the GNN block as shown in Figure 3. The GNN is followed by the UpdatePrefix module that is an MLP implementing the logic of selecting the appropriate side of the permutation to recurse on.

More formally, at each of the $O(\log m) = O(\log n)$ stages, each vertex in the GNN$^{\mathrm{mp}}$ maintains a list of which of its connecting edges are *active*. This requires $O(d)$ representation size. The goal next is to infer whether the number of connected components induced by the active edges is one, two, or more than two. This in turn decides the part of the ordering next stage will focus on. The computation of connected components can be carried out using at most two breadth first searches and hence via $O(D)$ rounds of a GNN$^{\mathrm{mp}}$ network. Following this intuition we arrive at the proposed architecture in Figure 3. Formally, we have the following guarantee.

**Theorem 9.** *For a fixed graph topology over $n$ nodes with maximum degree $d$ and diameter $D$, the network in Figure 3 produces the minimum cut. Furthermore, the network is of depth $\ell = O(D \log^2 n)$, size $O(n\ell)$, and has $O(d + \log n)$ parameters.*

*Proof.* Each vertex maintains an $O(d)$ sized representation indicating which of its edges are currently *active* plus additional constant number of values to indicate its component id during different runs of BFS. Given a list of active edges, the GNN module simply performs a procedure to compute whether the number of connected components is one, two, or more than two. This can be done with at most two BFS runs over the active edges. As we saw before in Section 3, this requires $O(D)$ depth.

At the end of the GNN module each vertex gets an integer value specifying its component id. The *UpdatePrefix* module then takes this information and is required to perform two computations: a) check if the number of connected components is one, two, or more than two. This requires checking the number of distinct elements in a list of $n$ numbers and can be done with an MLP $O(\log n)$ parameters and depth $O(\log n)$, b) update the set of active edges for each vertex depending on the number of connected components. This requires taking in the $O(d)$ sized representation and producing a new $O(d)$ sized representation for each vertex. This can be achieved again by an MLP using $O(d)$ parameters and depth $O(\log d)$. Once a given round of GNN and UpdatePrefix ends, the computations proceeds to the next layer. Importantly, the set of model parameters are shared across the different layers of the architecture as each time the computation required is the same. Hence overall we get $O(D \log n)$ depth and $O(d + \log^2 n)$ parameters. □

## 5 EXPERIMENTS

We show the efficacy of $\text{GNN}^+$ on the aforementioned graph problems: Shortest Paths, Effective Resistance, Affinity, MINCUT and MST, and compare to a state-of-the-art $\text{GNN}^{\text{mp}}$ model (Xu et al., 2019a).

**Dataset.** We generated synthetic random graphs between 500 and 1000 nodes. For the affinity measure, we used graphs with 250 nodes because of the need for using very dense graphs to have a reasonable number of alternate paths between any two end points. In general, we generated the data sets as follows: we fix the number of nodes $n$ in the graph to take values in $[250, 1000]$. For each value of $n$ we generate graphs from the Erdos-Renyi model $G(n, p)$ with edge sampling probability $p = \frac{\alpha}{n}$. We vary $\alpha$ depending on the problem. Specifically, we set $\alpha$ to be a constant in $[1, 100]$ to capture varying degrees of sparsity. For each $n, p$ we generate $30,000$ training examples consisting of tuples of the form $(G, s, t, d(s,t))$ where $G$ is a random graph drawn from $G(n, p)$, $s, t$ are two vertices uniformly drawn at random and $d(s,t)$ is one of shortest path value, effective resistance, or affinity between the two vertices. In the case of min cut and minimum spanning tree, we generate tuples $(g, v_G)$ where $v_G$ corresponds to the value of the minimum spanning tree or the global minimum cut.

**Models and Configurations.** For our baseline $\text{GNN}^{\text{mp}}$ implementation, we used the GIN model proposed in Xu et al. (2019a). This has been empirically shown (Xu et al., 2019a; Loukas, 2020; Errica et al., 2020) to be a state-of-the-art $\text{GNN}^{\text{mp}}$ model on several datasets. GIN updates feature representations $x_v^{(k)}$ of each node $v$ at iteration $k$ as: $x_v^{(k)} = \text{MLP}\Big( (1 + \epsilon^{(k)}) \cdot x_v^{(k-1)} + \sum_{u \in N(v)} x_u^{(k-1)} \Big)$, where MLP refers to a Multi-Layer Perceptron, $N(v)$ is the set of neighbors of $v$, and $\epsilon$ is a learnable parameter. For problems that involved weighted graphs (e.g. MST), we incorporated edge weights into the GIN update equation by replacing the sum of neighbor representations by a weighted sum.

Our $\text{GNN}^+$ implementation also used the same GIN implementation as its internal $\text{GNN}^{\text{mp}}$ block. All graphs in our experiments were undirected. For both baseline and $\text{GNN}^+$, we used node degree as the input node features for MINCUT and MST. For Shortest Paths, Effective Resistance and Affinity, we set input node features to be Booleans indicating if the node is a source/destination node or not.

Following Xu et al. (2019a), we performed 10-fold cross-validation for each of our experiments (corresponding to the two models and five problems), and report the average validation mean squared error(MSE) across the 10 folds. We run each 10-fold cross-validation experiment 10 times to compute confidence intervals. We apply batch normalization at each layer, and use an Adam optimizer and decay the learning rate by 0.5 every 50 epochs, and train for up to 600 epochs. For both the baseline $\text{GNN}^{\text{mp}}$ and the $\text{GNN}^+$ model, we tune the following parameters: initial learning rate $\in \{0.001, 0.003, 0.005, 0.007, 0.01, 0.03, 0.05\}$, number of hidden units $\in \{8, 16, 32, 64\}$, batch-size $\in \{32, 64\}$, dropout $\in \{0, 0.5\}$. For $\text{GNN}^{\text{mp}}$ we also tuned the depth (number of layers) $\in \{2, 4, 8, 12\}$. For the $\text{GNN}^+$ model, we tuned the number of parallel GNNs in each $\text{GNN}^+$ block to $\in \{1, 2, 3\}$ with GNN depth $\in \{2, 4\}$. We also tuned the number of $\text{GNN}^+$ layers $\in \{1, 2, 3\}$. We fixed the depth of each MLP block in $\text{GNN}^{\text{mp}}$ and $\text{GNN}^+$ to 2.

**Results.** To validate our theory regarding the better generalization bounds for $\text{GNN}^+$ models compared to $\text{GNN}^{\text{mp}}$, we compare the test mean squared errors for the two models across the five datasets. For all the five problem, Table 1 lists the test MSEs and corresponding standard deviations

| Problem | Label Variance | Avg. MSE (GNN$^{mp}$) | Avg. MSE (GNN$^+$) |
|---|---|---|---|
| Shortest Path | 7.482 | $0.975 \pm 0.031$ | $0.849 \pm 0.022$ |
| Effective Resistance | 7.949 | $0.397 \pm 0.025$ | $0.187 \pm 0.008$ |
| Affinity | 3.030 | $0.0025 \pm 1.75e{-}04$ | $0.0018 \pm 1.89e{-}05$ |
| MST | 4637.4 | $1011.39 \pm 106.94$ | $733.901 \pm 30.97$ |
| MINCUT | 11.964 | $0.963 \pm 0.110$ | $0.694 \pm 0.07$ |

Table 1: Performance of the GNN$^{mp}$ and GNN$^+$ architectures

for the two models. As a sanity check, we also plot the variance of the labels in our datasets, which corresponds to the MSE obtained by a naive model that predicts the mean label. We observe significant gains in accuracy of anywhere between 15% relative MSE improvement over the GNN$^{mp}$ baseline (for Shortest Paths) to as much as 108% relative MSE improvement (for Effective Resistance). Note that the naive mean predictor's MSE is at least an order of magnitude larger than all the MSE values for GNN$^{mp}$ and GNN$^+$ (except for the MSTdataset, where it is around five times larger - we suspect that the weighted graphs involved in this dataset make this a harder problem).

We posit that these accuracy gains directly stem from the sample-efficiency of the GNN$^+$ models as captured in Theorems 1,2 and 4 - the most compact GNN$^+$ networks that can represent these problems are smaller than the corresponding most compact GNN$^{mp}$ networks. Hence, by Theorem 5, such networks will have smaller generalization errors. In the appendix, we also plot the test accuracy as a function of number of epochs that suggest that our models also converge faster than the baseline GNN$^{mp}$ models, though we do not have any theoretical justification supporting this observation.

**Experiments on Real World Data.** We further demonstrate the applicability of our proposed GNN$^+$ architecture for solving classification tasks involving real world graphs. We experiment with the following real world datasets (Yanardag & Vishwanathan, 2015) that have been used in recent works for evaluating various GNN architectures (Xu et al., 2019a): 1) IMDB-BINARY and 2) IMDB-MULTI datasets: These are movie collaboration datasets with nodes as actors and the class label being the genre. 3) COLLAB: This is a scientific collaboration dataset with three classes. 4) PROTEINS: This is a bioinformatics dataset with 3 class labels. 5) PTC, 6) NCI1 and 7) MUTAG: These are various datasets of chemical compounds with two class labels each.

We train our GNN$^+$ proposed architecture on these graphs using the cross-entropy loss and as before compare with the GIN architecture of Xu et al. (2019a). We use the same input node features as in Xu et al. (2019a) and use the same experimental methodology as that for synthetic graphs above. In particular, when tuning hyperparameter tuning we allow the GNN$^{mp}$ architecture to explore depth upto 9 whereas the GNN$^+$ architecture is tuned by restricting the depth upto 3. The results are summarized in Table 2 below. As can be seen, in each instance GNN$^+$ either outperforms or matches the performance of the GNN$^{mp}$ architecture in terms of final test accuracy.

| Dataset | Test Acc. (GNN$^{mp}$) | Test Acc. (GNN$^+$) |
|---|---|---|
| IMDB-BINARY | $0.742 \pm 0.09$ | $0.769 \pm 0.02$ |
| IMDB-MULTI | $0.523 \pm 0.06$ | $0.527 \pm 0.04$ |
| COLLAB | $0.802 \pm 0.02$ | $0.816 \pm 0.004$ |
| PROTEINS | $0.7602 \pm 0.008$ | $0.7654 \pm 0.015$ |
| NCI1 | $0.849 \pm 0.004$ | $0.851 \pm 0.003$ |
| PTC | $0.686 \pm 0.02$ | $0.708 \pm 0.018$ |
| MUTAG | $0.876 \pm 0.016$ | $0.898 \pm 0.012$ |

Table 2: Performance of the GNN$^{mp}$ and GNN$^+$ architectures on real world classification datasets.

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
