# OpenReview forum: "Beyond GNNs: A Sample Efficient Architecture for Graph Problems"
_ICLR.cc/2021/Conference — Reject_

### Official Review · AnonReviewer1 · 2020-10-24
**Several parts unclear**

**Rating:** 4
**Confidence:** 3

**Review:**

Overview:
========
The paper suggests a modification to graph neural networks, which is claimed to overcome GNN expressiveness issues recently shown by Loukas (ICLR 2020).

Comments:
=========
I had multiple difficulties in following the content of the paper, some are detailed next.

-- "Unfortunately, [GNNs] require large depth as proved in Theorem 6 above"
The lower bound in Theorem 6 is for approximation alpha < 1.5, but here you discuss approximation O(log n), doesn't this void the lower bound? Why cannot usual GNNs produce an O(log n) approximation (and specifically Bourgain's embedding)?

-- "While there do exist semidefinite programming based algorithms (Linial et al., 1995) for computing the embeddings required for Bourgain’s theorem, they are not suitable for implementation via efficient neural architectures. Instead [...] we adapt the sketch based approximate shortest path algorithms of Das Sarma et al. (2010)"

In fact, the algorithm you implement is the one from Bourgain and Linial et al., not Dar Sarma et al. All those works use roughly the same sketching algorithm, which measures the distance of each point to randomly chosen clusters. However, the distance estimation procedure you implement (specifically ~d_G(s,t) = max_i|v_s^(i) - v_t^(i)|) is Bourgain's (this is just an ell-infinity embedding). Dar Sarma et al.'s estimation procedure is different and (building on the well-known work of Thorup-Zwick) relies on computing the common nearest neighbors of the given pair s,t in the random clusters. Note that this bears on the correctness of the proof of Theorem 8 (I think the statement still holds due to Matousek's analysis of Bourgain's embedding, but not for the reason you cite). Also note that in Theorem 8 (as well as all aforementioned results) c needs to be an integer (in particular, it is known that any approximation less than 3 is impossible with less than ~Omega(n^2) parameters).

-- For min-cut, you write that traditional GNNs require Omega(sqrt(n)) depth/rounds, citing Loukas (2020). But doesn't that lower bound entail both the depth and the width (d*sqrt(w) = ~Omega(sqrt(n)))?

-- I am unable to follow the proof of Theorem 9. Could you please explain the correctness of your construction.

(On this note, the sentence "Karger & Stein (1996) implies that with probability at least 1/n^2 there exists a prefix L' of L such that..." seems like an unfortunate inaccuracy; the "prefix" exists deterministically, and their guarantee is that the iterative random contraction algorithm finds it with probability at least ~1/n^2.)

Conclusion:
=========
I am currently unable to recommend accepting this paper, due to what seems like multiple inaccuracies, misinterpretations of prior work, unclear statements, and possibly technical correctness issues. I will await clarifications from the authors on the points detailed above.

Post discussion update
=========
After discussion with the authors, I have calibrated my score upward to 4, since the authors seemed willing to engage in discussion and correct/improve the paper, which I appreciate, but I still recommend not accepting the paper.

The authors generally acknowledged (though have not yet fixed) the issue of wrong attribution of the APSP algorithm. This isn't just a matter of citing B instead of A; the paper (still) contains a lengthy discussion of why A is not suitable, so instead they must resort to B, even though in reality they just use A (and B remains unused). This is glaring since these papers are famous classics, widely taught in graduate courses, their content is well known, and it is puzzling how a diametrically incorrect representation of them made its way into the paper.

The reason I dwell on this is that it signifies a larger issue with the paper. The original version was peppered with formal statements which were at best inaccurate, and even though the authors fixed (or said they would fix) the ones I pointed out, I remain unable to trust the overall technical soundness of the paper. The review time frame doesn't allow a reviewer to carefully verify every statement (nor would I want to), there must be some commitment of due diligence on part of the authors, that up to a small inevitable fraction of inaccuracies, the formal content is rigorously correct. I'm afraid the current version of the paper is quite off this mark.

Putting formal soundness aside, my present understanding of the idea of the paper is the following: The authors observe that many basic computations on graphs can be parallelized into a few computations of small width and depth. Usual GNNs can implicitly implement this if their width is large enough, but this poses a computational burden, and there are obvious advantages to explicitly building this parallelism into the architecture. This seems like a sensible and potentially empirically useful observation, but the experimental section still seems too thin to make the case properly. That said, perhaps I have not fully understood the paper, since its frequent inaccuracies and fuzzy statements made it a bit hard for me to follow.

In conclusion. I think the paper should undergo a substantial revision:
1. Clean up the theory part and ensure its formal soundness,
2. Crystalize the point of the paper (in particular, rather than just presenting GNN+, I hope a revised version would include a more thorough comparison with usual GNNs - not just dismiss them with some citations of prior works which allegedly prove limitations - this leaves doubts about the exact model and assumptions, particularly since as discussed above, the prior work is not always cited accurately),
3. Possibly expand the experimental section.

---

### Official Review · AnonReviewer3 · 2020-10-27
**Interesting theoretical observations but with several issues**

**Rating:** 5
**Confidence:** 3

**Review:**

The authors propose a variation of GNN which they name GNN+, which runs several GNN modules in parallel (with weight sharing). The authors show that it works well in practice on several synthetic datasets, and show many theoretical results.
I have some issue with the theoretical results and would raise the score if addressed.

- Unless I am mistaken the  GNN+ architecture requires different features inputs for all copies of the GNN, this should be added into the parameter counting that is done.
- When the parameters are the same across layers they are counted as one, while this is true if you enforce weight sharing but in general we train without this constraint. Furthermore this isn't done for GNNs which leads to an unfair comparison. The method should be the same, preferably without discounting weight sharing.
- Maybe I am missing something basic but the node affinity definition doesn't make any sense to me. A_p(u,v) should be just p times the indicator of (u,v)\in E so shouldn't be more informative then just the basic adjacency matrix.
- It doesn't seem like there is any theoretical gain when comparing theorem 6 and 8. Considering that in theorem 6 alpha<3/2 and the number of parameters is O(B^2L) (also, GNN don't have to have a fixed representation size each layer) where B*L>Omega(n). While in theorem 8 we get number of parameters O(n^(2+1/c)log(n)) which for c<2 where they agree doesn't seem to give a better bound.
- You do not show how to implement Bellmen Ford with a GNN which is missing from the proof.




Minor remarks:
- In App A first equation should be min not max
- In alg. 1 should be x_i=min(x_i,a_i+1) not x_i=min(x_i,a_i), also switch between x and v which is confusing.

---

### Official Review · AnonReviewer2 · 2020-10-31
**Interesting paper**

**Rating:** 8
**Confidence:** 3

**Review:**

The main question this paper tackles is: can one develop sample efficient architectures for graph problems while retaining the simplicity of the message passing framework?

While combining message passing with GNNs has shown to have positive empirical results,  we do not know of a general neural architecture that is  versatile enough to provably encode the solution space of a variety of graph problems such as shortest paths and minimum spanning trees.

This paper introduces a theoretically principled architecture -- GCN+ -- which is attempts to make GCNs more efficient by using ideas from the subfields of sketching approximations and parallel computing.

############

I vote for accepting the paper due to its novelty and the pros listed below.

############

Pros

+ An interesting paper with novel contribution combining ideas from parallel computing and sketching approximations.
+ Major algorithmic contribution of interest to practitioners
+ Theoretical contributions in the form of several theorems in the paper

Cons

- No code provided with the submission

---

### Official Review · AnonReviewer4 · 2020-11-03
**Promising GNN architecture but practical implementation and utility is unclear**

**Rating:** 5
**Confidence:** 3

**Review:**

This paper proposes a new building block for GNNs, called GNN$^+$. This building block trades of depth for width and involves multiple parallel regular GNN processing units. Using the GNN$^+$ architecture, the authors establish bounds for the required network depth (and total parameters) for several combinatorial problems over graphs.

Pros:
+ A parallel processing architecture, as opposed to sequential processing, is also likely to be computationally more efficient in large memory setups.

+ I haven't carefully checked all the proofs, but the key theorems seem relevant and interesting.

+ The paper is clearly written for most parts and the key intuitions behind the theoretical results are explained in sufficient depth.


Cons:
- The parallel GNN$^{mp}$ and MLP modules within a GNN$^+$ block all share parameters and hence, the only source of difference in the computed function is due to differences in initialization of node features. How is that difference executed in practice, both in the problems considered in this work and more generally for any graph problem? From the description in the experiments section, it seems that the networks are implemented using same set of features for the nodes during initialization (eg, node degrees, booleans for source and/or destination nodes).

- The paper lacks evidence if this model outperforms GNNs on real-world graph datasets. All the problems considered in this work are combinatorial optimization problems on Erdos-Renyi graphs, for which non-learning based solutions also exist. While many of the theoretical results are also custom to these problems, one would hope that the architecture is also evaluated on standard tasks (eg, classification) for real-world benchmark graph datasets.

- The central premise of the paper seems to be that the GNN$^+$ models require less depth (and total parameters) compared to standard GNN architectures. This is not clear from the experiments. A more descriptive result would be to assess the performance of these models as a function of depth (and total parameters).

---

### Decision · Program_Chairs · 2021-01-07
**Final Decision**

**Decision:**

Reject

**Comment:**

The paper presents a new GNN+ architecture and provide interesting theoretical observations about the architecture. The paper is quite promising and has several interesting insights. However, most of the reviewers believe that the paper is not ready for publication and can be significantly improved by: a) more formal and precise statements, b) clarifying the key points of the paper, c) more thorough experimental validation of the framework on real-world datasets.